# Disinfectants against SARS-CoV-2: A Review

**DOI:** 10.3390/v14081721

**Published:** 2022-08-04

**Authors:** Shuqi Xiao, Zhiming Yuan, Yi Huang

**Affiliations:** 1Wuhan Institute of Virology, Chinese Academy of Sciences, Wuhan 430020, China; 2National Biosafety Laboratory, Chinese Academy of Sciences, Wuhan 430020, China

**Keywords:** SARS-CoV-2, disinfectant, virucidal activity, alcohol, quaternary ammonium salt, chlorine-releasing agents, chlorine dioxide, hydrogen peroxide and peracetic acid, iodophor, ozone

## Abstract

The pandemic due to Severe Acute Respiratory Syndrome Coronavirus 2 (SARS-CoV-2) has emerged as a serious global public health issue. Besides the high transmission rate from individual to individual, indirect transmission from inanimate objects or surfaces poses a more significant threat. Since the start of the outbreak, the importance of respiratory protection, social distancing, and chemical disinfection to prevent the spread of the virus has been the prime focus for infection control. Health regulatory organizations have produced guidelines for the formulation and application of chemical disinfectants to manufacturing industries and the public. On the other hand, extensive literature on the virucidal efficacy testing of microbicides for SARS-CoV-2 has been published over the past year and a half. This review summarizes the studies on the most common chemical disinfectants and their virucidal efficacy against SARS-CoV-2, including the type and concentration of the chemical disinfectant, the formulation, the presence of excipients, the exposure time, and other critical factors that determine the effectiveness of chemical disinfectants. In this review, we also critically appraise these disinfectants and conduct a discussion on the role they can play in the COVID-19 pandemic.

## 1. Introduction

Since the first outbreak at the end of 2019, severe acute respiratory syndrome coronavirus 2 (SARS-CoV-2) is still raging around the world, bringing about detrimental effects to the world economy and society [1,2,3]. As of May 2022, there have been over 515 million confirmed cases of COVID-19, including more than 6 million deaths, reported by the World Health Organization (WHO) [4]. Current studies suggest that the SARS-CoV-2 virus can spread from an infected person’s mouth or nose in small liquid particles when they breathe, sneeze, cough, or speak [5]. It may also be transmitted via contact by touching contaminated surfaces, followed by touching the mouth, nose, or eyes. Experimental studies have shown that SARS-CoV-2 can survive on various plastic, latex, glass, and metal surfaces for hours to days [6]. Additionally, epidemiological evidence from the field suggests that the virus can survive on the outer packaging of cold-chain foods kept in a low-temperature environment and has been proven to maintain infectivity [7,8]. Therefore, the fomite transmission of SARS-CoV-2 is certainly plausible [9].

A highly effective treatment for this emerging infectious disease is lacking to date, although several drugs and vaccines have been found to improve clinical outcomes in large trials, and the rapid development and production of vaccines has permitted large-scale vaccination in many countries [10,11,12]. However, vaccine development still faces challenges, even with novel platforms [13]. More evidence is required before we know exactly how effective these drugs and vaccines are, especially when new virus variants constantly emerge [14,15]. These challenges become even greater due to the virus’s high transmissibility rate and long incubation period, as was evident with the Omicron variant [3]. In this context, preventive measures such as rapid detection, the isolation of cases, and the early quarantining of close contacts of positive cases, as well as mask use, physical distancing, hand hygiene, and surface disinfection, are crucial for reducing the risk of transmission. The use of chemical disinfectants has long been a widely accepted practice for infection prevention and control to protect healthcare professionals, patients, and people at a high risk of serious illness. For example, multi-user items (such as shopping carts, elevator buttons, door knobs, etc.) are considered high risk for transmitting the virus and require frequent decontamination with effective biocidal agents [16].

Besides the mode of action of the chemical disinfectant, the susceptibility of viruses to chemical disinfectants generally varies depending on their structure [17]. The other important factor is the environment, and viruses can remain infectious for several days to several months under different conditions [9,18,19]. On the other hand, the disinfectant formulations are quite complex and may include auxiliary substances such as surfactants or emollients in addition to active substances. The improper selection and inadequate use of sanitizers and disinfectants plays a significant role in the cross-transfer and spread of pathogens resulting in additional public health concerns [20]. More meaningful studies are needed to objectively evaluate disinfectants’ efficacy for suitable disinfectant selection and proper use.

The interest and demand for virucidal disinfectants have increased dramatically since the outbreak of COVID-19. There have been many studies that have evaluated the virucidal activity of disinfectants and disinfection methods against SARS-CoV-2, a novel coronavirus that is the infectious agent of the current COVID-19 pandemic. Here, we review the literature with regard to the inactivation of SARS-CoV-2 by microbicides intended for the decontamination of surfaces, for the decontamination of liquids, for hand hygiene, and oral rinses. Our discussion is limited to chemical microbicides and a general description of the mode of action for each class of chemical disinfectants, while their virucidal efficacy against SARS-CoV-2 is also presented. The stated purpose of this review is to provide information, primarily to healthcare facilities and laboratories, regarding a range of chemical disinfectants effective in mitigating SARS-CoV-2 transmission and pandemic control and identify knowledge gaps for virucidal efficacy against SARS-CoV-2. As such, information pertaining to surrogate viruses is not considered in this review. In addition, physical inactivation approaches (e.g., heating, ultraviolet radiation, and gamma irradiation) are outside of the scope of this review.

## 2. Alcohol-Based Disinfectants

Alcohols, namely ethanol and isopropanol, exhibit a broad spectrum of germicidal activity against bacteria, viruses, and fungi. Additionally, they have been used as low-level disinfectants in healthcare settings for many years [21]. Studies have shown that varied types and concentrations of alcohols inactivate SARS-CoV-2. As we summarized in Table 1, the effective concentration of alcohol disinfection is 30–95% and a contact time of several seconds or more is usually sufficient, which mostly results in a decrease of 3–4 log_10_. It is worth mentioning that this is the in vitro experimental testing time, but not the recommended time, for practical purposes.

Lipid membrane dissolution and protein denaturation are key mechanisms of the antimicrobial action of alcohol, leading to the disruption of the membrane and the inhibition of the metabolism [34]. Alcohols are amphiphilic compounds, as they possess both hydrophilic and lipophilic (hydrophobic) properties that facilitate their entry through the viral envelope. The outermost membrane of SARS-CoV-2 comprises lipids, and the antimicrobial mechanism of alcohol against SARS-CoV-2 and other enveloped viruses is similar to that for bacteria, since both have a lipid-rich outer membrane [34,35,36]. Due to its relatively greater lipophilicity, isopropanol is considered more effective than ethanol against SARS-CoV-2. Furthermore, recent studies have found that ethanol inhibits protein synthesis in *Escherichia coli* by its direct effects on ribosomes and RNA polymerase [37]. The efficacy of the alcohol-based disinfectants for inactivating SARS-CoV-2 depends on several key factors, which are outlined below:

Concentration: The optimum bactericidal concentrations of alcohols range from 60% to 90% *v*/*v* solutions in water but are generally ineffective against most microorganisms below 50% *v*/*v* [38]. The effect of different concentrations of alcohol against SARS-CoV-2 is shown in Table 1. Recent studies have shown that >30% concentrations of ethanol or isopropanol were effective in inactivating SARS-CoV-2 within 30 s [23,30,39], though several studies provided conflicting results [25,32]. This is due to the method for assessing the virucidal efficacy of a disinfectant in terms of factors such as the low susceptibility, low virus load, etc. [24,25].

Dirt and soil contamination: It is quite likely that the effect of a disinfectant is reduced in the presence of dirt or soil. Hand washing with soap coupled with an alcohol gel sanitizer was shown to be more effective than either agent used alone, with activity persisted for longer [40].

PH: Coronaviruses are reported to be more stable at a slightly more acidic rather than alkaline pH, with a high and low pH causing the inactivation of SARS-CoV [41,42]. The virucidal activity of ethanol against MS2 phage is significantly increased with the addition of sodium hydroxide due to protein denaturation [43]. Citric acid, malic acid, and urea (2%) have been reported to increase the effectiveness of alcohol-based sanitizers [44,45].

Excipients: Glycerin is usually added in hand sanitizers as a humectant, but its negative impact has been noted in several studies. For example, reducing the glycerol content in WHO-recommended formulations provided a better balance between antimicrobial efficacy and skin tolerance [23,46]. The removal of glycerol markedly increased the antimicrobial activity of isopropanol-based sanitizer through agglomerates of flaking skin cells forming in the sticky glycerol [23,47,48].

Owing to the increasing demand to control the spread of SARS-CoV-2, ethanol, isopropanol, and n-propanol are commonly applied as disinfectants, as summarized in Table 1. Additionally, alcohol-based hand sanitization is widely considered to be effective for reducing or eliminating the viral load. The most commonly used formulations for hand sanitizers are rinses, foams, gels, wipes, and sprays. Alcohol-based hand rubs in the form of foams, rinses, and gels did not differ significantly in trials of antimicrobial activity, but the application volume and drying time had a profound effect on their efficacy [49]. Gel-based hand sanitizers are reported to be more efficient against enveloped viruses, while foam-based preparations have the most rapid drying time [50]. Propanol has a marginally higher boiling point than ethanol, hence the drying time of isopropanol is slightly longer compared to ethanol [51]. The WHO has recommended two alcohol-based hand sanitizer formulations which are widely followed throughout the world, and both the original and modified WHO formulations have been shown to be effective against SARS-CoV-2 [23].

Alcohol disinfectants are not only used for skin disinfection, but also for inanimate surfaces such as stainless steel, plastic (PET), glass, PVC, and cardboard [27]. Moreover, they are easy to use and inexpensive, have non-toxic residue, and have acceptable odor and a rapid onset of action. In addition, alcohol is not significantly impaired by organic matter contamination [52]. However, alcohols that are flammable and explosive should be used with caution. Because alcohols are poor cleaners compared to other disinfectants and evaporate rapidly, they are not appropriate for use on environmental surfaces except those formulations containing alcohol plus other active agents such as quaternary ammonium or phenolic components. Additionally, alcohols are irritating to the eyes and skin, with long-term use damaging the skin [53]. Multiple disinfectants containing alcohol combined with other active agents such as quaternary ammonium or phenolic compounds are widely used for disinfecting environmental surfaces in healthcare facilities. However, it is worth noting that anionic additives in hand disinfectants containing alcohol may negate the efficacy of chlorhexidine gluconate persistence [39].

## 3. Quaternary Ammonium Salt Disinfectants

Quaternary ammonium compounds (QACs) are among the most commonly used disinfectants in healthcare and food-processing environments, as well as in the home. It was proposed that the following series of events is involved in the mechanism of action of QACs against microorganisms: (1) QACs’ adsorption to and penetration of the cell wall; (2) their reaction with the cytoplasmic membrane (lipid or protein), followed by membrane disorganization; (3) the leakage of intracellular lower-weight material; (4) the degradation of proteins and nucleic acids; and (5) cell wall lysis caused by autolytic enzymes [54]. Thus, QACs, as cationic detergents, are effective against bacteria, yeast, and lipid-containing viruses. QACs are also effective against non-lipid-containing viruses and spores, depending on the product formulation, because they interact with intracellular targets and bind to DNA [55].

Recent results have shown that QACs are effective at inactivating SARS-CoV-2 (Table 2) and are already the most widely represented class of disinfectants on the Environmental Protection Agency’s (EPA) List N [56]. However, the systematic evaluation for QACs or related products against SARS-CoV-2 is currently lacking, including their virucidal effects under different conditions according to the serial concentration, contact time, or different temperature. According to the results summarized in Table 2, we cannot compare and draw a conclusion on their virucidal activity against SARS-CoV-2 due to the different concentrations, exposure times formulations, etc. [57]. Moreover, relatively few studies have been specially conducted to assess the efficacy in practice.

Dilutable cleaner (alkyl (50% C14, 40% C12, 10% C16) dimethyl benzyl ammonium chloride, 2.9% *w*/*w*) can inactivate SARS-CoV, but not HCoV-229E, and there are no data for SARS-CoV-2. The disinfectant wipes containing the lower concentration of the same active ingredient (0.19% *w*/*w*) can inactivate other beta-coronaviruses (such as SARS-CoV), but not SARS-CoV-2 [59]. Therefore, tests for claims against a specific organism must be conducted with the specific organism to ensure its efficacy. Moreover, there are some divergences about whether QAC disinfectants work best against SARS-CoV-2. For instance, one review article reported that benzalkonium chloride was probably “less effective” against SARS-CoV-2, which was cited by the Centers for Disease Control (CDC) of the United States as a reason to avoid using benzalkonium chloride-based hand sanitizer products [16,65]. At the same time, the Environmental Protection Agency (EPA) of the United States and Health Canada both include benzalkonium chloride products on their official list of disinfectants recommended for use against SARS-CoV-2 [56]. Additionally, current data cannot indicate clearly whether benzalkonium chloride is effective against SARS-CoV-2 and its working condition (Table 2). However, benzalkonium chloride has several advantages: it is non-toxic, less irritating to the skin, and non-flammable. In particular, switching from alcohol to benzalkonium chloride hand sanitizer can lead to better hand hygiene compliance from healthcare workers, possibly decreasing the overall viral contamination of their hands [66]. However, more research is needed in this area.

The efficacy of QACs is dependent not only on the target organism but also on the method of application. Bolton et al. compared a hydraulic spray apparatus and a robotic wiping device for sanitizing surfaces [67]. It was found that the QAC was more effective than chlorine bleach in the spray apparatus but not in the robotic wiping device. The other important factor in the method of QAC application is to ensure the proper dosage. For example, the effective dose of the QACs can be compromised when combined with cotton mops and cleaning towels, because QAC concentrations can be reduced by 50% to 83% by cotton and microfiber cloths [68,69]. Moreover, the contact times for products containing alcohol plus other active agents vary considerably based on their content. In some cases, purified QACs were used rather than formulations designed for a specific organism or application, leading to generalized statements that QACs overall are not effective against the target organism [70], and the use of ethanol along with QACs has usually been associated with effective antimicrobial activity against coronaviruses [71]. All of the above emphasize that application methods have to be considered in the assessment of QAC activity. The proper concentrations and contact time as indicated on product labels should be used and monitored, and overdilution or overdose and insufficient contact time are critical factors that should be avoided. Using disposable disinfecting wipes or other ready-to-use products is an option to deliver an effective concentration of the QACs. Meanwhile, many other factors also have to be considered carefully, including the environment (e.g., liquid, surface, etc.), the presence of organic load, temperature, exposure time, and concentration, etc.

In addition, QACs are very diverse because of their wide range of chemical structures, which contributes to a continued increase in efficacy for specific applications and target organisms while lowering toxicity, and helps account for their widespread use [72]. These variations can affect the antimicrobial activity of the QACs in terms of dose and action against different microorganisms. For example, methyl group lengths of C12 to C16 usually show the greatest antimicrobial activity. Every QAC formulation has its advantages and disadvantages for a particular situation. The appropriate use of QACs can significantly reduce the number of infections.

Although the clear and severe threat posed by SARS-CoV-2 prompts a massive use of QACs to mitigate the spread of the infection, there are still concerns regarding the potential side effects of QACs on human health, animals, the environment, and the ecological balance. The improper selection and use of disinfectants plays a significant role in the cross-transfer and spread of pathogens resulting in additional public health and environmental concerns [73,74,75].

## 4. Chlorine-Releasing Agents and Chlorine Dioxide

Despite the introduction of many classes of disinfectants, disinfection approaches that liberate free available chlorine, such as hypochlorous acid and hypochlorite ions, continue to play an important role in improving public health by reducing the cross-transmission of infectious agents via drinking water and environmental surfaces. A large number of antimicrobial chlorine compounds are commercially available, including sodium and calcium hypochlorites, liquid chlorine, and inorganic and organic chloramines [76,77]. We performed a review of related studies to provide information on these chlorine-releasing agents and chlorine dioxide used for SARS-CoV-2 (Table 3). In these studies, the types of virucidal chlorine compounds examined comprised only a few common varieties, which helped these results guide the practical application of disinfectants and enabled easier standardization in laboratory assessments.

## 5. Sodium Hypochlorite

A variety of commercial products used in the home and healthcare facilities contain 1% to 15% sodium hypochlorite, with the most prevalent products being aqueous solutions of 4% to 6% sodium hypochlorite, which are usually called household bleach. The WHO recommends that regular household disinfectants containing 0.1% sodium hypochlorite (1000 mg/L) should be applied to various household surfaces [80]. The CDC recommends using 1/3 cup of bleach added to 1 gallon of water for surfaces exposed to COVID-19 patients, which is approximately 64 times diluted and has an available chlorine content of roughly 781 mg/L. A “strong chlorine solution” is a 0.5% solution of hypochlorite (containing approximately 5000 ppm free chlorine) used for disinfecting areas contaminated with body fluids, including large blood spills [17]. According to the data summarized in Table 3, this disinfectant could efficiently inactivate SARS-CoV-2 within 5 min.

## 6. Hypochlorous Acid

Hypochlorite produced by hypochlorite disinfectant can damage the lipids of the membrane and the nucleic acids due to its permeability through membranes and strong oxidizing ability. Moreover, it could inhibit the key enzymatic reactions within the cell and protein denaturation [36,77,81]. As the pH of the solution increases, the hypochlorite ion (-OCl) becomes predominant and the biocidal activity decreases [36]. In addition, organic matter and porous materials diminish the virucidal activity because of the quenching of free chlorine, though this chlorine-derived compound does exhibit significant efficacy against coronaviruses on non-porous surfaces [82].

The present results for SARS-CoV-2 were consistent with the mechanism of action (Table 3). The results of some studies showed the inefficient inactivation of SARS-CoV-2 because of the lower concentration or the shorter contact time than other studies [32,83]. On the other hand, some results effectively indicated the reduction in SARS-CoV-2 viability at low concentrations due to the lack of sufficient organic matrix during the tests [7,84]. It is worth noting that the results of some studies are not accurate yet, though they did not achieve a 4 log_10_ removal due to the high detection limit of inactivation caused by the cytotoxicity of disinfectants such as bleach (10%) and 84 disinfectant [25,79].

It is important to note that this inorganic hypochlorite disinfectant is only used on environmental surfaces and is not to be ingested. High concentrations of chlorine can lead to the corrosion of metal and the irritation of the skin or mucous membrane, in addition to potential side effects related to the smell of chlorine for vulnerable people such as asthmatics [83]. Excess chlorinated disinfectants take some time to degrade when they enter the natural environment and may exhibit acute toxicity to aquatic organisms [85]. However, hypochlorous acid is inexpensive, is generally nontoxic, and can be used within mouthwashes, sanitizers, and clinical disinfection at 1000 ppm, and as a part of wound care [84].

## 7. Chlorine Dioxide

Chlorine dioxide (ClO_2_), an alternative disinfectant to chlorine, has been widely used to control a number of waterborne pathogens in water and wastewater treatments [86]. It is an effective disinfectant in both liquid and gas states, making it a versatile biocidal agent [87,88]. For example, ClO_2_ can be safely used in low concentrations around animals and people to control airborne viruses [89]. Compared with chlorine, ClO_2_ is less toxic because of the greatly reduced generation of toxic halogenated disinfection products [90,91]. It is considered an alternative to chlorine.

The virucidal mechanism of ClO_2_ appears to be different for different types of viruses. One mode of action mainly involves the degradation of the viral proteins which are responsible for interactions with the host cell and injection mechanisms. Therefore, the attachment of the virus to host cells is inhibited, resulting in the inactivation of viruses. It has also been proposed that ClO_2_ can act on the viral genome. Specifically, the inactivation by ClO_2_ is caused by damage in the 5′ noncoding region within the genome, which is necessary for the formation of new virus particles within the host cell [92].

At present, there are several studies on the virucidal activity of ClO_2_ toward viruses including SARS-CoV-2. For instance, researchers achieved 5 logs of viral titer reduction using pure ClO_2_ at 80 ppm against SARS-CoV-2 in a suspension for as little as 10 s [78]. Another study followed the ASTM 2197-17 standard and showed that ClO_2_ at a lower concentration of 100 ppm did not fare as effectively against SARS-CoV-2 when dried on a hard non-porous surface, with only a 1.39 log_10_ reduction after a full 10 min of exposure; however, increasing the concentration to 500 ppm produced more favorable results [30]. The possible reasons for this discrepancy are that the study that inactivated SARS-CoV-2 at lower concentrations of ClO_2_ used a suspension test, reduced the protein content, used greater volumes of ClO_2_, and had relatively less virus. These factors illustrate the importance of comparing the efficiencies of biocides and their practical use under real-world conditions.

When SARS-CoV-2 viruses were treated with the same concentration of ClO_2_ or sodium hypochlorite (24 ppm), ClO_2_ reduced the viral titer to below the detection limit (≤2.2 log_10_ TCID_50_/mL) in 10 s in the presence of 0.5% FBS (fetal bovine serum) and by >4 log_10_ TCID_50_ in 30 s in the presence of 1.0% FBS. By contrast, 24 ppm of sodium hypochlorite inactivated only 99% or 90% SARS-CoV-2 in 3 min under similar conditions. This suggests that ClO_2_ is a much more powerful disinfectant than sodium hypochlorite, especially when organic matter is present in the contaminants.

In addition, it has also been demonstrated that ClO_2_ can denature proteins by the oxidative modification of tryptophan and tyrosine residues [93]. Various mutant strains of SARS-CoV-2 have a mutation in asparagine at position 501 to tyrosine (N501Y) in the spike protein, which is also responsible for receptor binding, and ClO_2_ might inactivate these novel mutants efficiently. Taken together, these observations might point to ClO_2_ being more useful than sodium hypochlorite for inactivating SARS-CoV-2.

Many factors have been found to exert great impacts on virus inactivation rates, including ClO_2_ dosage, pH, and temperature. The virus inactivation rates in ClO_2_ disinfection increase rapidly with increasing pH and temperature [94].

Overall, chlorine compound-based disinfectants have held a predominant position as reliable disinfectants because they have many of the properties of an ideal disinfectant, including a broad antimicrobial spectrum, rapid action, reasonable persistence in treated potable water, ease of use, solubility in water, relative stability both in its concentrated form and when diluted, relative nontoxicity to humans at use concentrations, a lack of poisonous residuals (reduced predominantly to chloride as a result of its oxidizing action of inorganic and organic compounds), its action as a deodorizer, being colorless, nonflammable, and nonstaining, in addition to having a low cost [95]. Moreover, chlorinated disinfectants can destruct viral nuclear acid by the formation of chloramines and nitrogen-centered radicals or the degradation of the 5′ noncoding region of the viral genome [94,96]. Wu et al. reported that chlorine disinfectant (trichloroisocyanuric acid) could destroy SARS-CoV-2 RNA after 2–3 h of exposure [58]. The disadvantages include the fact that it could cause irritation to mucous membranes; it has the potential to interact with some chemicals, resulting in the formation of toxic chlorine gas; there is an odor when it is used in concentrated forms; it has deleterious effects on some metals; and it has decreased efficacy in the presence of an organic load. Therefore, the cleaning and removal of organic matter before disinfection is recommended. In addition, a biocide’s pH and total chlorine availability have the greatest influence on biocidal efficacy. With the COVID-19 pandemic ongoing, there are limited available laboratory data on the efficacy of the chlorinated disinfectants of SARS-CoV-2. It is necessary to determine more precise times-to-inactivation and efficiencies that are used in practice under real-world conditions.

## 8. Hydrogen Peroxide and Peracetic Acid

Both hydrogen peroxide and peracetic acid are strong oxidizing agents and demonstrate broad-spectrum efficacy against a variety of microorganisms including bacteria, yeasts, and viruses [36].

### 8.1. Hydrogen Peroxide

Hydrogen peroxide is widely used for disinfection, sterilization, and antisepsis due to its ease of handling and expeditious start-up. It is considered environmentally friendly because it can rapidly degrade into innocuous products (water and oxygen) during dissolution, and is therefore a non-pollutant. It is also non-toxic, and is thus safe to use as a disinfectant for medical equipment and surfaces, even skin. Solutions in concentrations varying from 3% for routine disinfection to 25% for high level disinfection have been used [97]. However, the presence of catalase or other peroxidases in these organisms can increase the tolerance in the presence of lower concentrations [36]. Additionally, higher concentrations (10% to 30%) and longer contact times are required for sporicidal activity [98]. Not only can hydrogen peroxide be applied to surfaces in aqueous form, but it can also be used in vaporized form by a process called fumigation. Due to the ability of hydrogen peroxide vapor to decontaminate surfaces that are difficult to reach, it may be more beneficial for the decontamination of whole rooms, such as laboratories and patient rooms in hospitals. Furthermore, its biocidal activity is significantly increased in the gaseous phase [99]. Interestingly, a study found that hydrogen peroxide added to foam is more effective at higher temperatures when inactivating *Bacillus thuringiensis* spores compared to its liquid counterpart [100].

Hydrogen peroxide acts as an oxidant by producing hydroxyl free radicals, which react with lipids, proteins, nucleic acids, the cleavage of the RNA and DNA backbone, and oxidation, causing denaturation of proteins and the disruption of biological membranes and sulfhydryl bonds in proteins and enzymes. Due to their low molecular weight, hydrogen peroxide molecules can traverse through microbial cell walls and membranes to act intracellularly without having first induced cell lysis [99,101,102].

Studies have shown that hydrogen peroxide is virucidal (>4 log_10_ reduction) against FCV, adenovirus, AIV, and TGEV (as a SARS-CoV surrogate) at the lowest vaporized volume tested [103]. Additionally, a commercial product containing liquid hydrogen peroxide with surfactants was effective (>4 log_10_ TCID_50_/mL reduction) at a concentration of 0.5%, with an incubation time of 1 min against HCoV-229E [104]. Recent studies indicated that SARS-CoV-2 can be inactivated effectively by 0.1% hydrogen peroxide within 60 s of exposure on various surfaces [28]. However, a limitation to this study was that the hydrogen peroxide was examined on clean surfaces; therefore, further studies examining the impact of organic material and soil are necessary to determine its efficacy in a range of environments and situations. Hydrogen peroxide solutions (usually at the recommended oral rinse concentrations of 1.5% and 3.0%) showed weak viricidal activity after contact times of 15 s to 30 s, which were chosen to represent convenient, routinely achievable, and recommended time periods for oral rinsing in clinical setting (Table 4).

### 8.2. Peracetic Acid

Peroxyacetic acid is considered a more potent biocide than hydrogen peroxide against a broad spectrum of pathogens at lower concentrations (<0.3%) [105]; thus, it is frequently recommended for disinfecting medical devices [17]. However, higher concentrations of peracetic acid (>100 ppm) may be necessary to reduce non-enveloped viruses on surfaces, foods, and fomites [106].

Peroxyacetic acid also decomposes to safe by-products (acetic acid and oxygen) and has the added advantages of being free from decomposition by peroxidases, in contrast to hydrogen peroxide, and remaining active in the presence of organic loads. As with hydrogen peroxide, vapor-phase peroxyacetic acid is also more active (as oxidants) at lower concentrations than in the liquid form. Its main application is as a low-temperature liquid sterilant for medical devices, flexible scopes, etc., and is also used as an environmental surface sterilant [36]. Its main advantages over other vapor-phase systems include low toxicity, rapid action, and good activity at lower temperatures.

Similar to hydrogen peroxide, peroxyacetic acid probably denatures proteins and enzymes and increases cell wall permeability by disrupting sulfhydryl (-SH) and sulfur (S-S) bonds [36]. Finnegan et al. published an in vitro study on the action of hydrogen peroxide and peroxyacetic acid on proteins under physiological conditions. They found that peroxyacetic acid, in particular, oxidizes amino acids efficiently, degrades bovine serum albumin, and reduces the efficiency of the enzyme alkaline phosphatase at millimolar concentrations. These multiple targets imply that microbial organisms are less likely to mobilize resistance [99]. Additionally, there was an apparently large number of free radicals that arose from the reactions of the peroxide with organic compounds, and free radicals are highly reactive; peroxyacetic acid probably inhibits or kills microorganisms using several mechanisms, though the exact mechanism is still controversial due to the complexity of the reaction pathway [107].

Ansaldi et al. reported the effectiveness of peroxyacetic acid on coronaviruses: a 0.035% (35 ppm) solution inhibited SARS-CoV replication in a cell culture with a contact time of <2 min, while the same concentration did not affect the viral genome after 30 min of exposure [105]. Another study suggested that SARS-CoV can be inactivated with 500 to 1000 ppm of peroxyacetic acid [108]. A recent study showed similar results; the effective inactivation of SARS-CoV-2 after an exposure time of 60 s in carrier tests was documented by more than 4.0 log_10_ [28]. This study found that peroxyacetic acid could inactivate SARS-CoV-2 (4 log_10_ reductions) in 5 min in an ethanol bath at −20 °C, but could not completely destroy the RNA of SARS-CoV-2 after 3 h of exposure [58].

## 9. Iodophor

Iodophor is a complex of iodine and a solubilizing agent or carrier because iodine alone is not stable in water. This formation allows the sustained release of iodine and has powerful microbicidal activity [109]. The most commonly used iodophor is povidone iodine because of its rapid, broad-spectrum antimicrobial activity, even at low concentrations, and due to its established safety profile [110,111].

It is free molecular iodine that mediates the antimicrobial activity of iodophor. Iodine rapidly penetrates into microorganisms and reacts with key groups of proteins (in particular, the free sulfur amino acids cysteine and methionine leading to the loss of protein disulfide linkages) [17,81]. The iodination of phenolic and imidazole groups of the amino acids tyrosine and histidine and pyrimidine derivatives of cytosine and uracil lead to steric hindrances in hydrogen bonds and the denaturation of DNA. Iodine binding to unsaturated fatty acids has been shown to alter the physical properties of lipids and lipid-containing membranes which culminate in cell death [112].

As one of the important medicines on the WHO List of Essential Medicines, povidone-iodine (polyvinylpyrrolidone iodine, PVP-I) is routinely used in surgical procedures, including the disinfection of skin when formulated into scrubs or handwashes and for oral cavities through oral sprays and mouth rinses [113,114,115]. The combination of PVP-I with alcohol as a disinfectant shows excellent residual efficacy, and could reduce the amount of alcohol required, plus serve as a useful substitute or supplement to alcohol for disinfecting skin, oral cavities, and fomite surfaces.

Numerous studies have reported its microbicidal activity against bacteria, fungi, and viruses [110,111,112]. Additionally, studies have shown that povidone-iodine is able to deactivate SARS-CoV and MERS-CoV at concentrations of 0.23% to 7.5% with 15 and 60 s exposures, respectively [115,116,117]. The exposure of SARS-CoV-2 to PVP-I at a concentration of 0.5% to 10% resulted in similar results in the suspension test, in which the virus titer dropped below the levels of detection after 30 s (Table 5).

## 10. Ozone

Ozone, a naturally occurring configuration of three oxygen atoms, is a reliable, clean oxidizing agent with a powerful microbicidal effect against bacteria, viruses, fungi, and protozoa [123,124]. Because ozone can dissolve within solution or be applied in gaseous form, it has been used widely in recent decades. In the disinfection processes, ozone is used in its gaseous or aqueous form depending on the type of decontaminated surfaces. Ozone gas may be used for the disinfection of hospital rooms or transport vehicles, whereas dissolved ozone may be used in water treatment and food disinfection. For wastewater treatment, ozone is a substantial disinfectant that can enhance biological water quality in less time and at a lower concentration with higher efficacy [125]. However, the presence of organic matter may lead to the lower efficacy of decontamination [126]. Moreover, both forms of ozone must be administrated with caution to prevent harm to personnel when inhaled [127].

As a strong oxidizing agent, ozone reacts with the cytoplasmic membrane, thereby breaking lipid components at various bond sites, to inactivate microorganisms [128,129]. In the case of viruses, ozone damages viral capsids, hindering their infectivity to new cells by peroxidative reactions. Enveloped viruses such as coronaviruses might be more sensitive to ozone than non-enveloped viruses due to the interaction of ozone with the lipid layer envelopes [130].

One study showed that a high concentration of ozone (27.73 ppm) inactivated SARS-CoV in 4 min. The medium (17.82 ppm) and low (4.86 ppm) concentrations could also inactivate SARS-CoV with different speeds and efficacy [131]. Hudson et al. reported that the maximum anti-viral efficacy of ozone required a short period of high humidity (>90% relative humidity) after the attainment of the peak ozone gas concentration (20–25 ppm). Mouse coronavirus (MCoV) on different surfaces (glass, plastic, and stainless steel) and in the presence of biological fluids was inactivated by ozone by at least 3 log_10_ in the laboratory and in simulated field trials [132,133]. Here, we summarized the data of the virucidal activity of ozone water (not gas) against SARS-CoV-2 due to the different experimental methods with other chemical disinfectants (Table 6).

Hu et al. implied that an ozone concentration exceeding 18 mg/L could reduce vital SARS-CoV-2 to an undetectable level effectively within 1 min [134]. However, further studies are needed to evaluate the disinfection efficacy of ozone water in real-world conditions, such as the impact of organic material, different surfaces, etc. Martins et al. showed a 2 log_10_ reduction in the SARS-CoV-2 titer, but no reduction in genome quantification, upon 1 min exposure to ozone water [135]. Further testing, such as using higher ozone concentrations, may help develop the optimal concentration for the environmental disinfection of SARS-CoV-2. In addition, the results of Skowron et al. showed that ozone water improved the microbicidal efficiency of the disinfectant regardless of the disinfectant type, helped to reduce the use of disinfectant concentrations, and limited the increase in the microbial resistance to disinfectants [124].

Ozone water is eco-friendly, has microbicidal properties, and shows a synergistic effect of a biocidal action with other chemical disinfectants. Taken together, ozone water offers an inexpensive and feasible alternative for the routine control of the environmental spread of SARS-CoV-2.

## 11. Others

There are many studies on other types of disinfectants that have been tested for their virucidal activities against SARS-CoV-2 in detail, including formalin, chlorhexidine digluconate, anionic surfactant, and novel disinfectants such as calcium bicarbonate with a mesoscopic structure (CAC-717), etc. Additionally, many more commercial formulations have also been investigated.

Several chemical disinfectants among them could effectively reduce the SARS-CoV-2 virus at an appropriate concentration at a reasonable contact time, especially some formulations mixed with alcohols, quaternary ammonium salts, chlorine compounds, peroxide, iodine compounds, and aldehydes with different proportions. However, their properties may need to be confirmed further, for instance, whether they are harmful to humans, animals, and the environment. The results for the virucidal activity of these chemical disinfectants against SARS-CoV-2 are summarized in Table 7.

## 12. Discussion

The literature on SARS-CoV-2 virucidal efficacy is being continually updated, so the information presented in this review should be considered a snapshot taken at the present point in time. Even so, certain classes of microbicidal agents have displayed good virucidal efficacy against SARS-CoV-2, including alcohols, quaternary ammonium compounds (e.g., benzalkonium chloride), phenolics (e.g., para-chloro-meta-xylenol or PCMX), detergents (e.g., soap dish or soap liquid), organic acids (e.g., citric, lactic, and salicylic acids), and other lipid disrupting agents. The same is true for protein-denaturing agents (alcohols, phenolics, oxidizers, and organic acids) and genome-degrading agents such as alcohols and oxidizing agents. However, antimicrobial activity can be influenced by many factors such as the disinfectant used (e.g., type, formulation effects, and concentration), the presence of an organic load, synergy, exposure times, temperature, test method, etc.

The viricidal effects of various disinfectants at different concentrations could differ due to the above factors. An alcohol-based disinfectant is a typical example that has been proven to completely inactivate SARS-CoV-2, with the virucidal activity depending on the percentage concentrations of alcohol [22,23,32]. This is true for the virucidal activity of chlorine-releasing disinfectants and chlorine dioxide against SARS-CoV-2 [7,24,78], but organic matter and porous materials could greatly diminish their virucidal activities. Of the widely used biocidal agents in healthcare, the in vitro disinfection’s effectiveness evaluation showed that benzalkonium chloride and chlorhexidine gluconate were significantly inferior in disinfection effectiveness for SARS-CoV-2 compared to alcohol-based disinfectants. However, the disinfection effectiveness of benzalkonium chloride (0.2%) and chlorhexidine gluconate (1%) increased when compared with lower concentrations and during evaluation using the skin model, which suggests the potential effectiveness of the disinfectant on the skin [23]. This is because their disinfectant effect can last after application, in contrast to alcohol-based disinfectants.

Even the virucidal effect of the same active ingredient with a different product type could differ [59]. A synergistic effect could be attributed to different virucidal activity mechanisms. For example, some anionic surfactants’ additions exhibited a significant increase in the virucidal activity of alcohols against SARS-CoV-2; these included odium dodecylbenzenesulfonate and sodium laureth sulfate, which are commonly used in dish soaps and liquid soaps, though hand soap and dish soap solution hardly increased the reduction factor value [29]. This provides the ongoing global challenge with a very simple solution to enhance the disinfection efficiency to lessen the spread of SARS-CoV-2 from often-touched contaminated surfaces. In another study, the preparation of disinfectant solutions using ozone water improved the microbicidal efficiency of the tested disinfectants, including quaternary ammonium compounds, oxidizing agents, chlorine compounds, and iodine compounds. At the same time, using ozone water can help reduce the use of disinfectant concentrations and limit the increase in the microbial resistance to disinfectants [124]. Ozone water itself has shown good virucidal activity against SARS-CoV-2 [134].

The ability of SARS-CoV-2 to remain viable on different surfaces for days to weeks has been well documented. The use of disinfectant-impregnated wipes is one of the most efficient and prevalent methods for the decontamination of high-touch environmental surfaces and non-critical medical devices in hospitals and other situations because of their acceptable compliance and easy application. The addition of mechanical wiping using disinfectant wipes impregnated with ethanol and NaOCl rendered the SARS-CoV-2 virus inactive almost immediately, with no viral transfer from the used wipes to adjacent surfaces, which indicated that incorporating disinfectants is in agreement with other studies [30,143,144,145]. However, wipes made of an inappropriate material could interact with the adsorbed active ingredient, resulting in a lower or even abolished disinfectant efficacy [69]. Several information gaps have to be filled to complement the products’ user manual for the disinfectant, wipes, and the workflow, including material compatibility (the combination of the wipe and disinfectant), liquor ratio (wipe mass/disinfection solution volume), contact time (of the disinfectant and wipes), and storage time.

An increasing awareness of the role of contaminated environmental surfaces in the transmission of viruses has highlighted the need for effective methods for cleaning and disinfecting inanimate surfaces. On the other hand, the adequate disinfection of hands is also an important way to prevent the indirect transmission of SARS-CoV-2, especially during the pandemic era. Based on the review findings in the literature, the original formulations of WHO-recommended hand rubs seem to be less active against SARS-CoV-2 compared with modified formulations [32]. A possible reason for this is that glycerol, a humectant that is added to hand sanitizers to reduce the loss of skin moisture, can reduce the efficacy of isopropanol-based sanitizers through agglomerates of flaking skin cells forming in the sticky glycerol [48]. Other commercially available personal care products were all able to reduce SARS-CoV-2 titers. For instance, some hand hygiene liquids/gels containing chloroxylenol, citric acid, lactic acid, or salicylic acid were also effective in reducing SARS-CoV-2 titers (Table 7). However, further studies are clearly needed on the optimum design and delivery form of agents for the efficient hand decontamination of SARS-CoV-2.

Beside inanimate surface disinfection and hand sanitization, high viral loads in the oropharynx of a person infected with SARS-CoV-2 beg the consideration of proper oral hygiene. Iodine in the form of a tincture has been routinely used in surgical procedures, with numerous studies have validating its safety. PVP-I mouthwash is also included in the WHO R&D blueprint for experimental therapies against COVID-19. Oral rinses containing PVP-I could lead to a >4 log_10_ reduction in SARS-CoV-2 (Table 5). The action of hydrogen peroxide oral rinses is inferior to PVP-I, while chlorhexidine gluconate (oral and skin formulations) seems to provide suboptimal virucidal activity in suspension tests. Other antiseptic oral rinses containing benzalkonium and ethanol or other agents have also been shown to deactivate SARS-CoV-2 (Table 7). In summary, for oral rinses and skin cleansers, products containing PVP-I should be preferred as its action is rapid and efficient. Soap, surfactant, and alcohol-based hand sanitizers are all excellent alternatives for hand hygiene.

The use of disinfectants has long been a widely accepted part of infection prevention and control, but the disinfectant formulations are complex and may include auxiliary substances which can influence the effect of the disinfectant. Therefore, it is important to compare the efficacy of disinfectant products using the appropriate tests according to the standards of different countries and regions. However, traditional residual virus detection in inactivation validation studies uses CPE and TCID_50_/plaque assays, which have several limitations. For example, a reduction factor of >4 cannot be reached, so lengthening the incubation times and having a large quantity of the culture are necessary due to the low initial titer of the virus used for the inactivation effect or cytotoxicity of certain disinfectants. Chin et al. tested 0.1% benzalkonium chloride against SARS-CoV-2, and no infectious virus could be detected after 5 min of incubation at room temperature [7]. However, its reduction factor is about 3.8 because of its cytotoxic effects. In other studies, benzalkonium chloride (0.2% *w*/*w*) and its related oral rinse (Dequonal) significantly reduced SARS-CoV-2 infectivity to undetectable levels; although, the maximum possible inactivation level in these tests was only approximate 2–3 log_10_ [60,63]. The integrated cell culture real-time quantitative RT-PCR method is a more feasible strategy that we used to evaluate the virucidal activity of several disinfectants for SARS-CoV-2 and Ebola virus [22,146]. This method utilizes the host cell as an efficient tool to separate infectious and noninfectious viruses, because only viable viruses can inject their genome into the host cell for amplification. The cells were incubated for an optimized period to amplify the viruses, decrease the limit of quantitation, and improve the sensitivity of detection. This method made it possible to evaluate the virus being present at levels lower than the limit of detection of the TCID_50_/plaque assay performed in cells. Higher log inactivation values might be possible without limitations on the amount of the challenge virus that can be applied. The methods described in this study are easy to perform and can be adapted to validate the inactivation of viruses in various matrices.

Finally, it is worth noting the harmful impacts on human and animal health and the environment and ecological balance caused by the undue use of disinfectants, though disinfectants and sanitizers are essential preventive measures against the COVID-19 pandemic. For instance, chemical disinfectants used as highly concentrated, aerosolized, or atomized disinfectants can easily be inhaled or absorbed into the skin. Disinfectants may cause mucosal irritation, inflammation, swelling, and the ulceration of the upper and lower respiratory tract. A few chemicals are absorbed quickly through the mucosa of various organs and organ systems (e.g., the central nervous system and gastrointestinal tract) into the bloodstream [75,147,148]. The excessive use of disinfectants also poses a potential threat to other living beings and ecosystems. Some chemical disinfectants may gain entry into rivers and lakes, with aquatic ecosystems at a risk of contamination [148,149]. For example, chlorine disinfectants undergo reactions with the dissolved organic matter of surface water to produce disinfectant byproducts, which are highly toxic to aquatic flora and fauna [148].

## 13. Conclusions

The ongoing COVID-19 pandemic caused by SARS-CoV-2 has drawn broad attention and initiated widespread academic research on various decontamination measures for the environment and population. Fortunately, SARS-CoV-2 is susceptible to a variety of disinfectants as summarized in this review. However, this wide range also means that care must be taken to choose the best product for the particular use. An appropriate choice is best made by the virucidity evaluation, toxicity, materials compatibility, cost, etc. Better standardized tests for a virucidual activity assessment should be adopted. An environmental impact assessment of the escalating use of disinfectants is needed and clear and comprehensive guidelines for disinfectant application are necessary. Current advances and the generation of novel disinfectants against COVID-19 provide hope for the development of safe, effective, and convenient disinfectants that are affordable to all and accessible under diverse environments with a minimum risk to health and the environment. We hope to provide a bridge between interested scientists from different disciplines including chemistry, biology, public health, etc. By designing tailor-made disinfectants or advanced formulations, public health experts can expect to make a more accurate choice of disinfectants for decontamination in healthcare settings as part of infection prevention and control for emerging infectious diseases.

## Figures and Tables

**Table 1 viruses-14-01721-t001:** The virucidal activity of alcohol-based disinfectants against SARS-CoV-2.

Active Ingredient	Concentration	Production Type	Disinfection Phase	Contact Time	Reduction of Viral Infectivity (log_10_)	Reference
Ethanol	95% (*v*/*v*)	Disinfection solution	Suspension test	15 s–8 min	>4	[22]
	80% (*w*/*w*)	Disinfection solution	Suspension test, skin model	5 s–1 min	>4.50, >4.14	[23]
	75% (*v*/*v*)	Disinfection solution	Suspension test (with organic matrix)	30 s–5 min	≥4.75	[24]
		Disinfection solution	Suspension test	15 s–8 min	>4	[22]
		Disinfection solution	Suspension test	1 min, 5 min	≥1.83, ≥2.0	[25]
	70% (*v*/*v*)	Hand sanitizer gel	Suspension test	30 s	≥3.22	[26]
		Hand sanitizer foam	Suspension test	30 s	≥3.10	[26]
		Disinfection solution	Suspension test	15 s, 30 s	>4.33, >3.63	[27]
		Disinfection solution	Stainless steel, plastic (PET), glass, PVC, and cardboard carrier test	30 s	≥4.1, ≥4.1, ≥3.8, ≥4.0, ≥3.8	[28]
				1 min	≥5.0, ≥5.0, ≥4.7, ≥4.9, ≥4.7	
		Disinfection solution	Suspension test	5–30 min	>4.8	[7]
		Disinfection solution	PVC carrier test	1 min	>5	[29]
		Hand sanitizer	Suspension test	1 min, 5 min	≥2.5	[25]
	66.5% (*v*/*v*)	Disinfection solution	Stainless steel carrier test (with organic matrix)	30 s–10 min	5.12	[30]
		Wipe		0 s–5 min drying post-wiping	6.32	
	63% (*w*/*w*)	Disinfection solution	Suspension test (without and with organic matrix)	3 min	>5	[31]
	60% (*w*/*w*)	Disinfection solution	Suspension test, skin model	5 s–1 min	>4.50, >4.14	[23]
	60% (*v*/*v*)	Disinfection solution	Suspension test (with organic matrix)	30 s–5 min	≥4.75	[24]
	57% (*v*/*v*)	Disinfection solution	Suspension test	15 s–8 min	>4	[22]
	54% (*w*/*w*)	Disinfection solution	Suspension test (without and with organic matrix)	3 min	>5	[31]
	50% (*v*/*v*)	Disinfection solution	Suspension test (with organic matrix)	30 s–5 min	≥4.75	[24]
	45% (*w*/*w*)	Disinfection solution	Suspension test (without and with organic matrix)	3 min	>5	[31]
	40% (*w*/*w*)	Disinfection solution	Suspension test, skin model	5 s–1 min	>4.50, >4.14	[23]
	40% (*v*/*v*)	Disinfection solution	Suspension test (with organic matrix)	30 s–5 min	≥4.75	[24]
	38% (*v*/*v*)	Disinfection solution	Suspension test	15 s–8 min	≥4	[22]
	36% (*w*/*w*)	Disinfection solution	Suspension test (without and with organic matrix)	3 min	>5	[31]
	27% (*w*/*w*)/32.7% (*v*/*v*)	Disinfection solution		3 min	>5	
	30% (*v*/*v*)	Disinfection solution	Suspension test (with organic matrix)	30 s,1 min–5 min	4.42, ≥4.75	[24]
	20% (*v*/*v*)	Disinfection solution		30 s–5 min	1.08–1.92	
	20% (*w*/*w*)	Disinfection solution	Suspension test, skin model	5 s–1 min	0.08–0.81	[23]
	19% (*v*/*v*)	Disinfection solution	Suspension test	15 s–8 min	0.13–0.52	[22]
Isopropyl	70% (*w*/*w*)	Disinfection solution	Suspension test, skin model	5 s–1 min	>4.50, >4.14	[23]
		Disinfection solution	Stainless steel, plastic (PET), glass, PVC, and cardboard carrier test	30 s	≥4.1, ≥4.1, ≥3.8, ≥4.0, ≥3.8	[28]
				1 min	≥5.0, ≥5.0, ≥4.7, ≥4.9, ≥4.7	
		Disinfection solution	PVC carrier test	1 min	>5	[29]
Original WHO formulation I ^a^	100%	Hand rub formulations	Suspension test	1 min, 5 min	≥2.17, ≥2.25 ^#^	[25]
	40–80%		Suspension test (with organic matrix)	30 s	≥3.8	[32]
	30%			30 s	3.0	
Modified WHO formulation I ^b^	40–80%			30 s	≥5.9	
	30%			30 s	1.8	
Original WHO formulation II ^c^	30–80%			30 s	≥3.8	
Modified WHO formulation II ^d^	80%			30 s	5.3	
	30–60%			30 s	≥5.9	
Mikrozid^®^ universal ^e^	20%, 80%	Disinfection solution	Suspension test	15 s	≥4.02	[33]
Desmanol^®^ pure ^f^	20%	Hand sanitizer	Suspension test	15 s, 30 s	≥4.02, ≥3.02	
	80%			15 s, 30 s	≥2.02, ≥4.38	
Ethanol 35%, Isopropanol 35%	-	Disinfection solution	PVC carrier test	1 min	>6	[29]
Ethanol 35%, Isopropanol 35%, Glycerin 3%	-				>5	
Ethanol 70%, Sodium dodecylbenzenesulfonate 3%	-				>6	
Ethanol 70%, Sodium dodecylbenzenesulfonate 3%, Glycerin 3%	-				>6	
Isopropanol 70%, Sodium laureth sulfate 3%	-				>6	
Isopropanol 70%, Hand soap 3%	-				>7	
Ethanol 70%, Dish soap 3%	-				>7	
Ethanol 35%, Isopropanol 35%, Dish soap 3%, Glycerin 3%	-				>7	

^#^ Below the detection limit. ^a^ Original WHO formulation I consists of 80% (vol/vol) ethanol, 1.45% (vol/vol) glycerol, and 0.125% (vol/vol) hydrogen peroxide. ^b^ The modified WHO formulation I consists of 80% (wt/wt) ethanol, 0.725% (vol/vol) glycerol, and 0.125% (vol/vol) hydrogen peroxide. ^c^ Original WHO formulation II consists of 75% (vol/vol) 2-propanol, 1.45% (vol/vol) glycerol, and 0.125% (vol/vol) hydrogen peroxide. ^d^ The modified WHO formulation II contains: 75% (wt/wt) 2-propanol, 0.725% (vol/vol) glycerol, and 0.125% (vol/vol) hydrogen peroxide. ^e^ Mikrozid^®^ universal e; 100 g contains: 17.4 g propan-2-ol, 12.6 g ethanol (94%); ^f^ Desmanol^®^ pure: 100 g contains: 75 g propan-2-ol.

**Table 2 viruses-14-01721-t002:** The quaternary ammonium salt disinfectants against SARS-CoV-2.

Product/Active Ingredient	Production Type	Concentration	Disinfection Phase	Contact Time	Reduction of Viral Infectivity (log_10_)	Reference
Di-N-decyl dimethyl ammonium bromide (DNB)	Disinfection solution	≥283 mg/L	Suspension test (with organic matrix)	30 s–10 min	≥4.92	[24]
		212 mg/L		30 s, ≥1 min	3.59, ≥4.92	
		170 mg/L		30 s, ≥1 min	2.5, ≥4.92	
Di-N-decyl dimethyl ammonium chloride (DNC)	Disinfection solution	≥283 mg/L	Suspension test (with organic matrix)	30 s–10 min	≥4.92	
		212 mg/L		30 s, ≥1 min	3.59, ≥4.92	
		170 mg/L		30 s, ≥1 min	2.5, ≥4.92	
	Disinfection solution (mixed with ethanol as antifreeze)	3000 mg/L	Carrier test on cloth (−20 °C)	5 min	4	[58]
Benzalkonium chloride (BAC)	Disinfection solution	0.2%	Suspension test, skin model	5 s–1 min	1.83–3.19	[23]
		0.05%			1.33–2.36	
	Disinfection solution	0.1%	Suspension test	5 min–30 min	>3.8 ^#^	[7]
	Foaming handwash (0.1% *w*/*w*)	0.025%	Suspension test (with organic matrix) (37 °C)	1 min	≥3.4	[59]
	Disinfection solution (surface cleaner, 0.56% *w*/*w*)	0.45% *w*/*w*	Suspension test (with organic matrix)	5 min	≥4.5	
	Disinfection solution	0.2% *w*/*w*	Suspension test (with organic matrix)	15 s–30 s	2.09–>3.19 *	[60]
	Hand sanitizing wipe	0.13%		15 s–30 s	>2.64–>2.97 *	
Cavicide ^a^	Disinfection solution	-		15 s–30 s	>2.88–>3.19 *	
Clean Quick ^b^	Disinfection solution	diluted (0.02%)		15 s–30 s	0, >2.88 *	
MICRO-CHEM PLUS Detergent Disinfectant Cleaner ^c^	Disinfection solution	0.56–5%	Suspension test	15 s–8 min	>4	[22]
		0.19%		15 s, 30 s, >1 min	1.46, 3.23, >4	
		0.06%		15 s–4 min, 8 min	0–3.03, >4	
FWD ^d^	Disinfection solution	0.56–5%	Suspension test	15 s–8 min	>4	
		0.19%		15 s, >30 s	0.11, >4	
		0.06%		15 s–8 min	0	
Alkyl dimethyl benzyl ammonium chloride (C12-16) (0.096% *w*/*w*)	Disinfection solution (surface cleanser)	0.077% *w*/*w*	Suspension test (with organic matrix)	5 min	≥4.1	[59,61]
Alkyl (50% C14, 40% C12, 10% C16) dimethyl benzyl ammonium chloride (0.19% *w*/*w*)	Wipe	0.19% *w*/*w* (as supplied)	Carrier test on glass surface	2 min	≥3.5	
Disinfectant spray ^e^	Disinfection solution	-	Carrier test on glass surface	2 min	≥4.5	
RTU cleaner ^f^	Disinfection solution	0.092% *w*/*w* (as supplied)	Carrier test on glass surface	2 min	≥4.0	[59]
Super Sani-Cloth wipes ^g^	Wipe	-	Wipe glass surface	2 min	>4.46	[62]
Dequonal (Dequalinium chloride, benzalkonium chloride)	Oral rinses	-	Suspension test (with organic matrix)	30 s	≥2.61	[63]
Colgate Plax^®^ Fruity Fresh (0.075% Cetylperidinium chloride, 0.05% Sodium fluoride)	Oral rinses	-	Suspension test (without and with organic matrix)	30 s, 1 min	5	[64]
Mikrozid^®^ sensitive ^h^	Disinfection solution	20%	Suspension test	15 s, 1 min	≥4.02, ≥3.17	[33]
		80%		15 s, 30 s, 1 min	≥4.38, ≥4.38, ≥2.17	

* For these tests, the amount of inactivation detected was the maximum possible inactivation level the assay was able to detect. Variation in log reduction value for these data points is due to variation of the titer on different test dates, not variation in the inactivation activity of the disinfectant. ^#^ Below the detection limit. ^a^ Cavicide (Metrex Research LLC, Orange, CA, USA): Diisobutylphenoxyethoxyethyl dimethyl benzyl ammonium chloride (0.28%), isopropanol (17.20%). ^b^ Clean Quick (Procter & Gamble Company, Cincinnati, OH, USA): Alkyl dimethyl benzyl ammonium chlorides (0.15%), alkyl dimethyl ethylbenzyl ammonium chlorides (0.15%). ^c^ MICRO-CHEM PLUS Detergent Disinfectant Cleaner (MCP, National Chemical Laboratories, Inc., Philadelphia, PA, USA): 4-Nonylphenol, branched, ethoxylated 1–5%, Sodium Carbonate 1–5%, Alkyl (68% C12, 32% C14) dimethyl ethylbenzyl ammonium chloride 1–3%, Alkyl Dimethyl Benzyl Ammonium Chloride (C12–C18) 1–3%, with components not listed either non-hazardous or below reportable limits. Tested at 0.06–5% of supplied. ^d^ FWD: Similar to MCP but more environmentally friendly, FWD is also a dual quaternary ammonium compounds product which is still in the stage of research and development. Tested at 0.06–5% of supplied. ^e^ Disinfectant spray: 50% *w*/*w* ethanol, 0.083% *w*/*w* Alkyl (50% C14, 40% C12, 10% C16) dimethyl benzyl ammonium saccharinate. ^f^ RTU cleaner: Alkyl (67% C12, 25% C14, 7% C16, 1% C8–C10–C18) dimethyl benzyl ammonium chloride; Alkyl (50% C14, 40% C12, 10% C16) dimethyl benzyl ammonium chloride. ^g^ Super Sani-Cloth wipes: composed of two quaternary ammonium compounds (each 0.25% by weight) and isopropyl alcohol (55.5% by weight). ^h^ Mikrozid^®^ sensitive: 100 g contains: 0.26 g alkyl(C12-16) dimethylbenzyl ammonium chloride (ADBAC/BKC (C12e16)), 0.26 g didecyldimethyl ammonium chloride (DDAC), 0.26 g alkyl(C12e14)ethylbenzyl ammonium chloride (ADEBAC(C12e14)).

**Table 3 viruses-14-01721-t003:** The virucidal activity of chlorine-releasing disinfectants and chlorine dioxide against SARS-CoV-2.

Product/Active Ingredient	Production Type	Concentration	Disinfection Phase	Contact Time	Reduction of Viral Infectivity (log_10_)	Reference
Trichloroisocyanuric acid (TA)	Disinfection solution (mixed with ethylene glycol as antifreeze)	≥1000 mg/L	Carrier test on cloth (−20 °C)	5 min	4	[58]
	Disinfection solution	250 mg/L	Suspension test (with organic matrix)	5 min, 10 min, 20 min	3.25, 4.0, ≥4.75	[24]
		500 mg/L		30 s, ≥5 min (5 min, 10 min, 20 min)	3.58, ≥4.75	
		1000 mg/L		30 s, 5 min, 10 min, 20 min	≥4.75	
Sodium hypochlorite (NaOCl)	Disinfection solution	0.5% (*v*/*v*)	Carrier test on stainless steel (with organic matrix)	30 s, 1 min	2.03, 3.45	[30]
				5 min, 10 min	>4 *	
	Wipe			0–5 min drying post-wiping	>4 *	
	Disinfection solution	80 ppm	Suspension test	10 s–3 min	>4	[78]
		8 ppm		10 s–3 min	2–3	
		0.8 ppm		10 s–3 min	1	
		80 ppm	Suspension test (with organic matrix)	10 s–3 min	>4	
		8 ppm		10 s–3 min	1	
		0.8 ppm		10 s–3 min	1	
84 disinfectant (NaOCl)	Disinfection solution	600 mg/L	Suspension test	5–30 min	>3 ^#^	[79]
		500 mg/L		5 min, 10–30 min	2–3, >3 ^#^	
		400 mg/L		5–10 min, 15–30 min	2–3, >3 ^#^	
		300 mg/L		5–30 min	1–2	
Bleach (NaOCl)	Disinfection solution	10%	Suspension test	1 min, 5 min	≥3.25	[25]
Household Bleach	Disinfection solution	1:49 (~150 ppm)	Suspension test	5 min, 10 min, 30 min	>4.8	[7]
		1:99 (~75 ppm)	Suspension test	5 min, 10 min, 30 min	>4.8	
Sodium hypochlorite and hypochlorous acid	Disinfection solution	0.002% and 0.013%	Suspension test	1 min, 5 min	2.3, 3.75	[25]
Dilutable cleaner (Sodium hypochlorite)	Disinfection solution	0.14% *w*/*w*	Suspension test	30 s	≥5.1	[59]
		0.32% *w*/*w*	Suspension test (with organic matrix)	5 min	≥5.1	
Chlorine dioxide (ClO_2_)	Disinfection solution	100 ppm	Carrier test on stainless steel (with organic matrix)	30 s, 1 min, 5 min, 10 min	<1.15	[30]
		500 ppm		30 s, 1 min	2.07, 2.53	
				5 min	>4 **	
				10 min	>4 *	
	Wipe	100 ppm		0 s drying post-wiping	2.78 ***	
		500 ppm			4.27 ***	
Cleverin ^a^	Disinfection solution	80 ppm	Suspension test	10 s–3 min	>4	[78]
		24 ppm		10 s–3 min	>4	
		8 ppm		10 s–3 min	3–4	
		0.8 ppm		10 s–3 min	1	
		80 ppm	Suspension test (with organic matrix)	10 s–3 min	>4	
		24 ppm		10 s–3 min	3–4 ^a^	
		8 ppm		10 s–3 min	2–3	
		0.8 ppm		10 s–3 min	1	
Chlorine dioxide ^b^	Disinfection solution	80 ppm	Suspension test	10 s–3 min	>4	
		8 ppm		10 s–3 min	>4	
		80 ppm	Suspension test (with organic matrix)	10 s–3 min	>4	
		8 ppm		10 s–3 min	~2	

^a^ Cleverin (Taiko Pharmaceutical Co., Ltd., Osaka, Japan) is a mixture of 500 ppm ClO_2_, 17,900 ppm sodium chlorite, 3300 ppm decaglycerol monolaurate, and 80 ppm silicone. When the viruses were treated with 24 ppm ClO_2_ in the presence of 1.0% FBS (fetal bovine serum), the viral titer was decreased by about 4 log_10_ TCID_50_ even in 10 s. ^b^ Pure ClO_2_: ClO_2_ gas is dissolved in ultrapure water. * No viable virus remained. ** No detectable virus remained on the carrier surface in TCID_50_ assays; however, one of nine biological replicates (i.e., a single carrier from only one of three independent experiments) showed CPE in safety tests at the 5 min mark. *** Considerably high amounts of viable virus were recovered from both test (0 s, 30 s, 60 s, 5 min drying post-wiping) and transfer carriers, indicating that transferring an infectious material from one surface to another via wiping can occur. ^#^ Below the detection limit.

**Table 4 viruses-14-01721-t004:** The virucidal activity of peroxide-based disinfectants against SARS-CoV-2.

Product/Active Ingredient	Production Type	Concentration	Disinfection Phase	Contact Time	Reduction of Viral Infectivity (log_10_)	Reference
Hydrogen peroxide (H_2_O_2_)	Disinfection solution (oral rinse)	1.5% (*w*/*w*)	Suspension test	15 s, 30 s	1.33, 1.0	[27]
		3.0% (*w*/*w*)		15 s, 30 s	1.0, 1.8	
	Disinfection solution	0.1%	Stainless steel, plastic (PET), glass, PVC, and cardboard carrier test	30 s	2.4, -, 2.3, 2.4, -	[28]
				1 min	≥4.8, -, ≥4.5, ≥4.7, -	
Cavex Oral Pre-Rinse (Hydrogen peroxide)	Disinfection solution (oral rinse)	- *	Suspension test (with organic matrix)	30 s	0.33–0.78	[63]
Peracetic acid (PA)	Disinfection solution (mixed with ethylene glycol as antifreeze)	2000 mg/L	Carrier test on cloth (−20 °C)	>5 min	4	[58]
Oxivir Tb wipes ^a^	Wipe	-	Carrier test on stainless steel, laminate wood, and porcelain	2 min	>4.46, >4.37, >4.73	[62]

^a^ Oxivir Tb wipes are hydrogen peroxide-based (≥0.1% to <1% by weight) and benzyl alcohol-based (1–5% by weight). * The exact formulations for these oral rinses are not publicly available due to patent-related restrictions.

**Table 5 viruses-14-01721-t005:** The virucidal activity of iodophor against SARS-CoV-2.

Product/Active Ingredient	Production Type	Concentration	Disinfection Phase	Contact Time	Reduction of Viral Infectivity (log_10_)	Reference
Povidone-iodine (PVP-I)	Oral rinse	0.5%, 1.25%, 1.5%	Suspension test	15 s	>4.33	[27]
				30 s	>3.63 ^#^	
		0.5%, 0.75%, 1.5%		15 s	3.0 ^#^	[118,119]
				30 s	3.33 ^#^	
Povidone-iodine (PVP-I)	Oral rinse	0.125–0.25 mg/mL	Suspension test	30 s, 1 min, 2 min, 5 min	<2	[120]
		0.5 mg/mL		1 min	3.8 *	
		0.5 mg/mL		2 min, 5 min	>4	
		1 mg/mL, 2 mg/mL		1 min, 2 min, 5 min	>4	
Iso-Betadine mouthwash (Polyvidone-iodine)	Oral rinse	1%	Suspension test	30 s	≥2.61 ^#^	[63]
Povidone-iodine (PVP-I)	Disinfection solution	7.5%	Suspension test	5 min, 15 min, 30 min	>3.8 ^#^	[7]
Povidone-iodine (PVP-I)	Disinfectant solution	10%	Suspension test (with organic matrix)	30 s	≥4	[121]
	Throat spray	0.45%			≥4	
	Skin cleanser	7.5%			≥4	
	Gargle/mouthwash	1.0%, 0.5%			≥4	
Clyraguard ^a^	Disinfectant solution	undiluted	Suspension test	30 s, 10 min, 30 min, 60 min	0.07, 1.73, >3.47, >3.47	[122]

^#^ Below the detection limit. * No viable virus remained. ^a^ Clyraguard copper iodine complex, developed by Clyra Medical Technologies, Inc. Westminster, CA, USA, is a novel FDA-registered product intended to be used for decontaminating non-critical PPE. The formula has proven antimicrobial activity, and has been cleared for use on skin and wounds.

**Table 6 viruses-14-01721-t006:** The virucidal activity of ozone against SARS-CoV-2.

Product/Active Ingredient	Production Type	Concentration	Disinfection Phase	Contact Time	Reduction of Viral Infectivity (log_10_)	Reference
Ozone water	Disinfection solution	18, 36 mg/L	Suspension test	1 min	>3 *	[134]
Ozone water	Disinfection solution	0.2–0.8 mg/L	Suspension test	1 min	2	[135]

* No viable virus remained.

**Table 7 viruses-14-01721-t007:** The virucidal activity of other chemical disinfectants against SARS-CoV-2.

Product/Active Ingredient	Production Type	Concentration	Disinfection Phase	Contact Time	Reduction of Viral Infectivity (log_10_)	Reference
Formalin	Disinfectant solution	10%	Suspension test	1 min, 5 min	≥1.25	[25]
Virkon (21.41% Potassium peroxymonosulfate, 1.5% sodium chloride)	Disinfectant solution	2%	Suspension test	1 min, 5 min	≥3.0	
p-chloro-m-xylenol (PCMX)	Disinfectant solution	0.125% *w*/*w*	Suspension test (with organic matrix)	1 min	≥5	[59]
	Bar soap	0.014% *w*/*w*		30 s	≥4.1	
	Hand hygiene liquid	0.021% *w*/*v*		5 min	≥4.1	
Lactic acid	Disinfectant solution	1.9%		5 min	≥5.5	
Citric acid (1.9% *w*/*w*), lactic acid (0.51% *w*/*w*)	Hand sanitizer gel	1.5% *w*/*w* citric acid, 0.41% *w*/*w* lactic acid	Suspension test	30 s	≥4.7	
Citric acid (2.4% *w*/*w*)	Disinfectant wipes	2.4%	Carrier test on stainless steel (with organic matrix)	30 s	≥3.0	
Salicylic acid	Liquid gel handwash	0.025% *w*/*w*	Suspension test (with organic matrix)	30 s	≥3.6	
	Foaming handwash	0.023% *w*/*w*		30 s	≥5.0	
Hydrochloric acid	Disinfectant solution	0.25% *w*/*w*	Suspension test (with organic matrix)	30 s	≥4.1	
Chlorhexidine gluconate	Disinfection solution	0.2%	Suspension test, skin model	5 s–1 min	0.33–2.42	[23]
		1.0% *w*/*w*			1.0–3.17	
Potassium monopersulfate (KMPS)	Disinfectant solution	1%	Stainless steel carrier test (with organic matrix)	30 s, 1 min	2.54, 3.52	[30]
				≥5 min	>4	
	Wipe			0 s–5 min drying post-wiping	>5	
W30 (N-Alkylaminopropyl Glycine)	Disinfectant solution	0.25%, 0.5%	Suspension test	8 min	>4	[22]
		1%		≥2 min	>4	
CAC-717 (Calcium bicarbonate with a mesoscopic structure) ^a^	Disinfectant solution	-	Suspension test	2 s–60 min	>4 **	[136]
		1%, 2%, 10%	Suspension test	15 s–5 min	>4	[137]
		-	Suspension test (with organic matrix)	5 min	4.3	
Virusend (TX-10)	Disinfectant solution	-	Suspension test	1 min, 10 min	>4	[138]
			Carrier test on stainless steel	1 min, 10 min	>4.3	
AWC	Antimicrobial skin and wound cleanser	-	Suspension test	30 s, 1 min, 10 min	>3.5 ^#^	[139]
			Porcine skin test	30 s, 1 min, 10 min	2	
			Disinfectant coating tests on plastic	10 min, 1 h	≥2, ≥2.2	
			Disinfectant coating tests on porcine skin	10 min, 30 min, 1 h	>1	
AWC2			Suspension test	30 s, 1 min, 10 min	>3.5 ^#^	
			Porcine skin test	30 s, 1 min, 10 min	3	
			Disinfectant coating tests on plastic	10 min, 1 h	≥2.5 ^#^	
			Disinfectant coating tests on porcine skin	10 min, 30 min, 1 h	≥2	
Sodium laureth sulfate (SLS)	Household cleaning agents	0.1%	Stainless steel, plastic (PET), glass, PVC, and cotton fabric carrier test	30 s	3.1, ≥3.6, ≥3.3, ≥3.5, ≥3.1 ^#^	[28]
				1 min	≥4.9, ≥4.9, ≥4.6, ≥4.8, ≥4.4	
Liquid hand soap (Biodegradable amphoteric surfactants, DMDM hydantoin)	Household cleaning agents	-	Suspension test	1 min, 5 min	≥2.0, ≥2.25	[25]
Handwash (Sodium laureth sulfate, cocamidopropyl betaine)	Household cleaning agents	-	Suspension test	1 min, 5 min	≥0.83, ≥0.92	
Handwash (Chloroxylenol, PCMX)	Household cleaning agents	-	Suspension test	1 min, 5 min	≥0.83, ≥0.92	
Hand soap solution	Hand sanitizer	1:49	Suspension test	5 min, 15 min, 30 min	4.2, >4.8 ^#^	[7]
Rosin soap ^c^	Disinfectant solution	2.5% (*w*/*v*)	Suspension test	5 min	<2 ^#^	[140]
Thymol^®^ Mouthwash by Xepa (0.05% Thymol)	Oral rinses	-	Suspension test (without and with organic matrix)	30 s, 1 min	0.5–0.75	[64]
Bactidol^®^ (0.1% Hexetidine 9% Ethanol)	Oral rinses	-	Suspension test (without and with organic matrix)	30 s, 1 min	5.0	
Salt water (2% (0.34 M) Sodium chloride)	Oral rinses	-	Suspension test (without and with organic matrix)	30 s, 1 min	0	
Oradex^®^ (0.12% chlorhexidine digluconate)	Oral rinses	-	Suspension test (without and with organic matrix)	30 s, 1 min	4	
Chlorhexamed fluid (0.1% Chlorhexidine bis-(D-gluconate))	Oral rinses	80%	Suspension test	5 min, 10 min	0.37, 0.76	[141]
Chlorhexamed forte alkoholfrei (0.2% Chlorhexidine bis-(D-gluconate))	Oral rinses	80%	Suspension test	1 min, 5 min	0.4, 0.81	
Octenisept ^d^	Oral rinses	20%, 80%	Suspension test	15 s, 30 s, 1 min	≥4.38	
Chlorhexamed Forte (Chlorhexidinebis (D-gluconate))	Oral rinses	-	Suspension test with organic matrix	30 s	~1	[63]
Dequonal (Dequalinium chloride, benzalkonium chloride)	Oral rinses	-		30 s	~3	
Dynexidine Forte 0.2% (Chlorhexidinebis (D-gluconate))	Oral rinses	-		30 s	~0.5	
Listerine Cool Mint (Ethanol, essential oils)	Oral rinses	-		30 s	~3	
Octenident mouthwash (Octenidine dihydrochloride)	Oral rinses	-		30 s	~1	
ProntOral mouthwash (Polyaminopropyl biguanide, polyhexanide)	Oral rinses	-		30 s	~1.6	
ColdZyme^®^ (CZ-MD) ^e^	Mouth spray	-	Suspension test	20 min	1.76	[142]

** A reduction in viral titer of ≥4 log_10_ relative to treatment with maintenance medium. ^#^ Below the detection limit. ^a^ Has a pH of about 12.4 and contains calcium hydrogen carbonate particles (1120 mg/L) and carbon complex microparticles (50–500 nm). ^b^ BIAKōS antimicrobial skin and wound cleanser. The ingredients are: (i) polyhexamethylene biguanide (PHMB), a cationic antimicrobial that may attract and adhere to the negatively charged lipid layer, thereby inactivating the virus; (ii) vicinal diols (octane-1-2-diol and ethylhexylglycerin), which are capable of disrupting lipid structures; (iii) ethylenediamine tetracetic acid (EDTA), which is known to have antimicrobial activity that can synergize with various antimicrobials; (iv) poloxamer 407, a non-ionic surfactant that helps to solubilize lipids in water and maintains the activity of PHMB and vicinal diols. AWC is water-based, AWC2 uses ethanol as a vehicle. ^c^ Rosin soap was produced from crude tall oil by Forchem Ltd. (Rauma, Finland). It is a water solution obtained from dried rosin salt, consisting of less than 10% sodium salts of tall oil fatty acids and over 90% sodium salts of resin acids. The resin acids and fatty acids of the product originated from the coniferous trees *Pinus sylvestris* L. and *Picea abies* L. The most abundant resin acid types included abietic acid, dehydroabietic acid, pimaric acid, and palustris acid. ^d^ Octenisept; 100 g contains: 0.1 g octenidine dihydrochloride, 2 g phenoxyethanol. ^e^ CZ-MD solution contains glycerol, water, cod trypsin, ethanol, calcium chloride, tris, and menthol.

## Data Availability

All the data are publicly available and cited in in accordance with journal guideline.

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
