# Peer review of "Disinfectants against SARS-CoV-2: A Review"

_viruses, 2022, doi:10.3390/v14081721_

Round 1
Reviewer 1 Report
Firstly I would like to congratulate the authors - they have conducted an extensive review of the literature in an area which, as they point out, is continually expanding.
Unfortunately, the number of papers included in their review has (in my opinion) resulted in a loss of focus. A review article as currently titled "The disinfectants against SARS-CoV-2" would be of interest to many and of great use. However, the focus in many places of the manuscript, is the action of the disinfectant itself rather than its action or efficacy against SARS-CoV-2. It is stated in the introduction that efficacy against surrogate viruses is out of scope but surrogate data is presented/discussed as is efficacy against other viruses (and bacteria).
I would suggest significantly condensing this review and narrowing its focus to SARS-CoV-2 (as I believe was intended). This will also provide a good opportunity to highlight knowledge gaps and the problems associated with efficacy testing and data interpretation. For example, it would seem that most of the studies cited have reported results of suspension tests. How do these differ from surface tests - have enough studies been carried out/published to draw conclusions?
The authors do touch briefly on the potential impact of soil - this (and other variables) could/should be expanded upon. Do studies incorporate soil? What is the inoculum level - is this appropriate? How many studies demonstrate appropriate and effective neutralisation of the test disinfectant?
Finally, on the basis of current evidence, are the authors able to make recommendations regarding the efficacy of disinfectants against SARS-CoV-2 - what, if any, additional research is required?
Author Response
Response to Reviewer 1 Comments
Firstly I would like to congratulate the authors - they have conducted an extensive review of the literature in an area which, as they point out, is continually expanding.
Thanks.
Unfortunately, the number of papers included in their review has (in my opinion) resulted in a loss of focus. A review article as currently titled "The disinfectants against SARS-CoV-2" would be of interest to many and of great use. However, the focus in many places of the manuscript, is the action of the disinfectant itself rather than its action or efficacy against SARS-CoV-2. It is stated in the introduction that efficacy against surrogate viruses is out of scope but surrogate data is presented/discussed as is efficacy against other viruses (and bacteria).
The efficacy of disinfectants against SARS-CoV-2 is shown in several tables in our manuscript clearly, and we think that it is not necessary to make more additional descriptions in the text. As we stated in the manuscript, the information presented in this review should be considered a snapshot taken at the present point in time and more research is needed in this area, such as the action of some disinfectants against SARS-CoV-2, which are not clear until to now according to our review of the related literatures. We presented some surrogate data because of the limited research in this area and many researchers had cited some surrogate data for the same reason (Viana Martins et al., 2022). However, one of the stated purposes of this review is to identify knowledge gaps for virucidal efficacy against SARS-CoV-2. The focus of this review is to provide information in detail to healthcare facilities and laboratories, regarding a range of chemical disinfectants effective in mitigating SARS-CoV-2 transmission and pandemic control, including the virucidal efficacy data and the possible action. We hope to provide a bridge between interested scientists from different disciplines including chemistry, biology, public health, etc. By designing tailor-made disinfectants or advanced formulations, public health experts can expect more and accurate choice of disinfectants for decontamination in healthcare settings as part of infection prevention and control for emerging infectious diseases.
Eggers M, Schwebke I, Suchomel M, Fotheringham V, Gebel J, Meyer B, Morace G, Roedger HJ, Roques C, Visa P, Steinhauer K. The European tiered approach for virucidal efficacy testing - rationale for rapidly selecting disinfectants against emerging and re-emerging viral diseases. Euro Surveill. 2021 Jan;26(3):2000708. doi: 10.2807/1560-7917.ES.2021.26.3.2000708.
I would suggest significantly condensing this review and narrowing its focus to SARS-CoV-2 (as I believe was intended). This will also provide a good opportunity to highlight knowledge gaps and the problems associated with efficacy testing and data interpretation. For example, it would seem that most of the studies cited have reported results of suspension tests. How do these differ from surface tests - have enough studies been carried out/published to draw conclusions?
Thank you. There have been many studies evaluated the virucidal activity of disinfectants and disinfection methods against SARS-CoV-2 and there are some literatures focused on the virucidal activity of some type of disinfectants or disinfection methods against SARS-CoV-2 (Singh et al., 2020; Viana Martins et al., 2022). However, there is limited review to summary the virucidal activities of most common used chemical disinfectants against SARS-CoV-2. We reviewed the literature with regard to inactivation of SARS-CoV-2 by microbicides intended for decontamination of surfaces, for decontamination of liquids, for hand hygiene, and oral rinses and made a conclusion that SARS-CoV-2 is susceptible to a variety of disinfectants as summarized in this review. At the same time, this wide range also means that care must be taken to choose the best product for the particular use; an appropriate choice is best made by virucidity evaluation, toxicity, materials compatibility and cost etc; better standardized tests for virucidual activity assessment should be adopted; clear and comprehensive guidelines for disinfectant application are necessary. We discussed these knowledge gaps and the problems associated with efficacy testing and data interpretation in each part for different disinfectant and discussion section. Yes, the most of the studies cited have reported results of suspension tests. The suspension test is different with surface tests because these tests should be done according to the different standards (Eggers et al., 2021), and we have noted the different disinfection phase in our manuscript (Table 1-7). We need more testing results to draw conclusion including suspension tests and surface tests. However, there are no enough studies carried out and the related data is limited, so we suggested that more research about practical use under real-world conditions should be determined.
Singh D, Joshi K, Samuel A, Patra J, Mahindroo N. Alcohol-based hand sanitisers as first line of defence against SARS-CoV-2: a review of biology, chemistry and formulations. Epidemiol Infect. 2020 Sep 29;148:e229. doi: 10.1017/S0950268820002319.
Viana Martins CP, Xavier CSF, Cobrado L. Disinfection methods against SARS-CoV-2: a systematic review. J Hosp Infect. 2022 Jan;119:84-117. doi: 10.1016/j.jhin.2021.07.014.
Eggers M, Schwebke I, Suchomel M, Fotheringham V, Gebel J, Meyer B, Morace G, Roedger HJ, Roques C, Visa P, Steinhauer K. The European tiered approach for virucidal efficacy testing - rationale for rapidly selecting disinfectants against emerging and re-emerging viral diseases. Euro Surveill. 2021 Jan;26(3):2000708. doi: 10.2807/1560-7917.ES.2021.26.3.2000708.
The authors do touch briefly on the potential impact of soil - this (and other variables) could/should be expanded upon. Do studies incorporate soil? What is the inoculum level - is this appropriate? How many studies demonstrate appropriate and effective neutralisation of the test disinfectant?
The “soil” here means the different testing condition (generally clean or dirty condition) according to the test standards (Viana Martins et al., 2022). Dirty test condition means some organic matrix in disinfectant solution during the testing because the effect of disinfectant is possibly reduced in the presence of dirt or soil. We have presented the disinfection phase in Table 1-7, such as suspension test (without organic matrix), suspension test (with organic matrix), carrier test on …, etc. We did not summary the appropriate and effective neutralization of the test disinfectant because neutralization or neutralization agents for different disinfectant is different. The related information could be summarized in another review and we did not expand to review and discuss related information to avoid the length of this review to be too long and to avoid a loss of focus.
Finally, on the basis of current evidence, are the authors able to make recommendations regarding the efficacy of disinfectants against SARS-CoV-2 - what, if any, additional research is required?
As we known, virucidal activity can be influenced by many factors such as disinfectant used (e.g. type, formulation effects and concentration), presence of an organic load, synergy, exposure times, temperature, etc. We could not make simple recommendations according the data we summarized in this review because of the different concentration, exposure time and formulation, etc. And relatively few studies have been conducted to assess efficacy in practice especially. However, we discussed the action mode, advantages and disadvantages, problems remained, and make some recommendation in each type of disinfection part.
For example, quaternary ammonium compounds (QACs):
Recent results have shown that QACs are effective at inactivating SARS-CoV-2, and series of events involved in the mechanism of action of QACs against microorganisms were summarized in QACs part. There are still some divergences about whether QAC disinfectants work best against SARS-CoV-2, and current data can’t indicate clearly whether benzalkonium chloride is effective against SARS-CoV-2 and its working condition. However, benzalkonium chloride has several advantages: it is non-toxic, less irritating to skin, and non-flammable. In particular, switching from alcohol to benzalkonium chloride hand sanitizer can lead to better hand hygiene compliance from healthcare workers, possibly decreasing overall viral contamination of their hands. More research is needed in this area.
Reviewer 2 Report
Good work on the thorough literature search and compiling the data, valuable for the ongoing fight against Covid pandemic.
You will have to (re)write for easy reading and understanding, maybe worthwhile using a professional writer. I worked on the first couple of pages as a sample with few suggestions.

Author Response
Response to Reviewer 2 Comments
Good work on the thorough literature search and compiling the data, valuable for the ongoing fight against COVID pandemic.
You will have to (re)write for easy reading and understanding, maybe worthwhile using a professional writer. I worked on the first couple pages as a sample with few suggestions.
Thank you for your comments. We have made some revision according to your suggestion and then asked for editing service to (re)write our manuscript for easy reading and understanding. The responses to some suggestions are following:
- The last sentence of abstract: we have rewritten the sentence.
- The third paragraph of introduction:As we known, antimicrobial activity can be influenced by many factors such as disinfectant used (e.g. type, formulation effects and concentration), presence of an organic load, synergy, exposure times, and temperature etc. Many detailed information about the virucidal activities of disinfectants against SARS-CoV-2 are still unclear, and care must be taken to choose the best product for the particular use, so more meaningful studies are needed to evaluate disinfectants’ efficacy objectively.
- The first paragraph in “Alcohols based disinfectants”: We just want to note that the collected data in Table 1 is the in vitro experimental testing results as reference, not our recommendation on exposure time for practical purposes. The suitable exposure time and concentration should be determined according to the practical purposes and the evaluation of the virucidal effects in certain real-world application environment.
- The second paragraph in “Alcohols based disinfectants”: We have added reference.
- The third paragraph in “Alcohols based disinfectants”: cited one study and say several studies. Yes, we have added several references in revised manuscript. And “it” (in sentence “It is due to the …) means the conflicting results above.
- The fourth paragraph in “Alcohols based disinfectants”: The activity in the last sentence means the antimicrobial activity of soap hand wash coupled with alcohol gel sanitizer.

Reviewer 3 Report
The manuscript presented by the authors is relevant since it makes a rigorous description of the disinfection methods available for use against SARS-CoV-2. The emerging disinfection methods in the current COVID-19 pandemic become relevant as they are methods that impact the containment of the transmission of the virus.
Allow me to make some observations that could improve the quality of the document presented.
Introduction section:
- Since the first outbreak at the end of 2019, the severe acute respiratory syndrome coronavirus 2 (SARS-CoV-2) is still raging around the world, causing a detrimental effect on the world economy and society. (MISSING PLACE THE REFERENCE).
-line 35 “And field epidemiological evidence suggests that the virus can survive”….”survive” is correct? “infectious activity can be another synonym”
-These challenges become even greater due to its high transmissibility rate and a long incubation period, such as Omicron variant…LACK REFERENCE
-Line 84: SARS-CoV-2, is incorrect.
-Table 1 shows a lot of information that looks repetitive, I suggest that the content of the columns "Production type and Disinfection phase" be "merged" because there is information that is repeated.
-Line 121: Corona virus is incorrect (without space)
-This argument (line 260-261 remove). “We performed a review of related studies to provide information on chlorine releasing agents and chlorine dioxide used for SARS-CoV-2 (Table 3)”. Place another argument that allows linking the previous idea.
-Table 3. The virucidal activity of chlorine-releasing disinfectants against SARS-CoV-2. There is information that is constantly repeated, I suggest that you "merge" information.
-Line 281: The US CDC change by “The CDC”.
-Line 306-307: It is important to note that this inorganic hypochlorite disinfectant is only used on environmental surfaces and is not to be ingested into the human body change by “It is important to note that this inorganic hypochlorite disinfectant is only used on environmental surfaces and is not to be ingested”
-Line 343: The acronym FBS is not described anywhere in the manuscript.
-Consider the inclusion of additional information in the “hydrogen peroxide” section, the hydrogen peroxide plasma technology. There is information about its disinfectant activity on ESKAPE and SARS-CoV-2 bacteria.
-Line 438: by disrupting sulfhydryl -SH), midified by “(-SH)”
-Line 479: “Numerous studies have reported its microbicidal activity against bacteria, fungi, and viruses” lack references.
Author Response
Response to Reviewer 3 Comments
The manuscript presented by the authors is relevant since it makes a rigorous description of the disinfection methods available for use against SARS-CoV-2. The emerging disinfection methods in the current COVID-19 pandemic become relevant as they are methods that impact the containment of the transmission of the virus.
Allow me to make some observations that could improve the quality of the document presented.
Introduction section:
- Since the first outbreak at the end of 2019, the severe acute respiratory syndrome coronavirus 2 (SARS-CoV-2) is still raging around the world, causing a detrimental effect on the world economy and society. (MISSING PLACE THE REFERENCE).
We have added the reference here in revised manuscript.
-line 35 “And field epidemiological evidence suggests that the virus can survive”….”survive” is correct? “infectious activity can be another synonym”
Yes, we think that “survive” is correct and the similar description was used in some archives, such as van Doremalen et al., 2020 (van Doremalen, N., Bushmaker, T., Morris, D.H., Holbrook, M.G., Gamble, A., Williamson, B.N., Tamin, A., Harcourt, J.L., Thornburg, N.J., Gerber, S.I., Lloyd-Smith, J.O., de Wit, E., Munster, V.J. Aerosol and surface stability of SARSCoV-2 as compared with SARS-CoV-1. N. Engl. J. Med. 2020; 382, 1564–1567). It has the same meaning with the following description “has been proven to maintain infectivity”.
-These challenges become even greater due to its high transmissibility rate and a long incubation period, such as Omicron variant…LACK REFERENCE
We have added the reference here in revised manuscript.
-Line 84: SARS-CoV-2, is incorrect.
We think it is right.
-Table 1 shows a lot of information that looks repetitive, I suggest that the content of the columns "Production type and Disinfection phase" be "merged" because there is information that is repeated.
Though it looks repetitive in the columns of Production type and Disinfection phase, the two columns represent the different information. Production type containing solution, hand sanitizer gel and hand sanitizer foam etc. While Disinfection phase means that different approach for virucidal activity testing according different standards (Eggers et al., 2021).
Eggers M, Schwebke I, Suchomel M, Fotheringham V, Gebel J, Meyer B, Morace G, Roedger HJ, Roques C, Visa P, Steinhauer K. The European tiered approach for virucidal efficacy testing - rationale for rapidly selecting disinfectants against emerging and re-emerging viral diseases. Euro Surveill. 2021 Jan;26(3):2000708. doi: 10.2807/1560-7917.ES.2021.26.3.2000708.
-Line 121: Corona virus is incorrect (without space)
Yes, we have revised it in manuscript.
-This argument (line 260-261 remove). “We performed a review of related studies to provide information on chlorine releasing agents and chlorine dioxide used for SARS-CoV-2 (Table 3)”. Place another argument that allows linking the previous idea.
We have revised it in our manuscript.
-Table 3. The virucidal activity of chlorine-releasing disinfectants against SARS-CoV-2. There is information that is constantly repeated, I suggest that you "merge" information.
Similar with Table 1, we think each column contains necessary information for virucidal activity evaluation according different standards (Eggers et al., 2021).
-Line 281: The US CDC change by “The CDC”.
Yes, we have revised it.
-Line 306-307: It is important to note that this inorganic hypochlorite disinfectant is only used on environmental surfaces and is not to be ingested into the human body change by “It is important to note that this inorganic hypochlorite disinfectant is only used on environmental surfaces and is not to be ingested”
Thank you, we have revised it.
-Line 343: The acronym FBS is not described anywhere in the manuscript.
Yes, we have added the full name for FBS in manuscript: fetal bovine serum
-Consider the inclusion of additional information in the “hydrogen peroxide” section, the hydrogen peroxide plasma technology. There is information about its disinfectant activity on ESKAPE and SARS-CoV-2 bacteria.
Yes, there is much information about the disinfectant activity of hydrogen peroxide plasma technology on a variety of microorganisms. But we did not expand to review and discuss related information to avoid the length of this review to be too long.
-Line 438: by disrupting sulfhydryl -SH), midified by “(-SH)”
Yes, we have revised it.
-Line 479: “Numerous studies have reported its microbicidal activity against bacteria, fungi, and viruses” lack references.
We have added the reference here in revised manuscript.

Round 2
Reviewer 3 Report
The recommendations have been addressed correctly. Therefore I have no objection to accepting the manuscript for publication.